# Ultrasound-Guided Tru-Cut Biopsy in Gynecological and Non-Gynecological Pelvic Masses: A Single-Center Experience

**DOI:** 10.3390/jcm11092534

**Published:** 2022-04-30

**Authors:** Francesca Buonomo, Sofia Bussolaro, Clarice de Almeida Fiorillo, Danilo Oliveira de Souza, Fabiola Giudici, Federico Romano, Andrea Romano, Giuseppe Ricci

**Affiliations:** 1Institute for Maternal and Child Health, I.R.C.C.S. “Burlo Garofolo”, 34137 Trieste, Italy; clarice.dealmeidafiorillo@burlo.trieste.it (C.d.A.F.); federico.romano@burlo.trieste.it (F.R.); giuseppe.ricci@burlo.trieste.it (G.R.); 2Department of Medical, Surgery and Health Sciences, University of Trieste, 34127 Trieste, Italy; sofia.bussolaro@burlo.trieste.it (S.B.); fabiola.giudici@gustaveroussy.fr (F.G.); 3ELETTRA Sincrotrone Trieste S.C.p.A., S.S. 14 Km 163.5, 34149 Trieste, Italy; danilo.oliveiradesouza@elettra.eu; 4Department of Cardiac, Thoracic, Vascular Sciences and Public Health, University of Padova, 35122 Padova, Italy; 5S.C. (UCO) Anatomia ed Istologia Patologica, Azienda Sanitaria Universitaria Giuliano Isontina, 34148 Trieste, Italy; andrea.romano@asuigi.sanita.fvg.it

**Keywords:** tru-cut biopsy, ultrasound, gynecological oncological diseases

## Abstract

Aim: The aim of this study was to evaluate the feasibility of adequacy, accuracy, and safety of ultrasound-guided tru-cut biopsy in managing malignant and benign abdominopelvic masses in a selected population and critically discuss some issues in different situations, which deserve some reflections on those practices. Materials and Methods: This is a retrospective study involving 42 patients who underwent transvaginal or transabdominal tru-cut biopsy between August 2017 and November 2021. The inclusion criteria were poor health status or primary inoperable advanced tumor, suspicion of recurrence or metastasis to the ovaries or peritoneum in gynecological and non-gynecological pelvic malignancies. Tissue samples were considered adequate if it was possible to determine the origin of the tumor, and immunohistochemistry could be performed. Diagnostic accuracy was assessed considering the agreement between tru-cut biopsy histology and final postoperative histology. Results: It total, 44 biopsies were obtained from 42 patients (2 patients had repeat biopsies). The pathologist considered all pathological samples adequate (adequacy 100%). The final histology was consistent with tru-cut biopsy diagnosis in all but 2 cases (diagnostic accuracy 88.2%). If we consider only the cases that have carried out at least two diagnostic samples, accuracy rose to 94.1%. Pathological examinations from tru-cut samples showed 2 benign lesions (4.8%) and 40 malignant tumors (95.2%), divided into 19 advanced primary inoperable ovarian cancers, 7 primary advanced cervical cancers, 4 recurrent endometrial cancers, 3 recurrent cervical cancers, 3 recurrent ovarian cancers, 1 case of primitive peritoneal malignancy (leiomyosarcoma), and 3 non-gynecological cancers with a strong suspicion of metastases at ultrasound (2 cases of ovarian, colorectal cancer metastasis, and 1 case of pelvic site type B lymphoma metastasis). However, one case of minor complication related to the procedure was reported but not significant. Conclusions: The diagnostic adequacy, accuracy of the tru-cut biopsy, and safety were high. Pathological samples are representative of the disease and suitable for histological and immunohistochemical analysis.

## 1. Introduction

In the management of pelvic tumors, ultrasound (US) examination performed by an experienced operator is currently considered crucial to characterize the benign or malignant nature of the lesion, the site, the extent inside the pelvic, and/or abdominal cavity, providing helpful information for clinical management [1]. However, as already stated, the pathological diagnosis is fundamental to leading us to the best therapeutic choice. Minimally invasive biopsy methods can represent a valid alternative to major surgery and are therefore preferred in patients unsuitable for surgery or in the presence of suspected relapse [1].

In this context, we can mention the fine needle aspiration cytology (FNAC) procedure which involves only cytological evaluation, while fine needle aspiration biopsy (FNAB) provides tissue samples that are not always architecturally preservedv [2]. US-guided tru-cut needle biopsy (TCNB) detects specimens with preserved tissue architecture, allowing comprehensive histological evaluation, including immunohistochemistry [3]. For instance, this technique is a vital part of the prostate cancer diagnostic algorithm and the triple breast cancer evaluation [4,5]. The TCNB targets a malignant representative portion of a suspicious mass and is considered a simple procedure to obtain a tissue sample because it does not require any special patient preparation or general anesthesia [3]. Moreover, it is a low-cost technique and can be performed in an outpatient setting [6]. TCNB is considered a superior technique to the fine needle aspiration procedure [7]. Vlasak et al. compared the results of the ascites puncture with US-guided TCNB [8], finding significant differences in favor of TCNB for all the analyzed variables (malignancy confirmation 72.9% vs. 95.8%, tumor origin 52.1% vs. 89.6%, histologic subtype 43.8% vs. 85.4% and accuracy 43.7% vs. 95.4%). However, evacuative paracentesis improved the patient’s subjective relief. Since biopsy samples are larger than in FNAB, the most architecturally preserved tissue is involved in TCNB (93–100% vs. 72–92%, respectively [9]. These characteristics make the sample adequate and suitable for immunohistochemical study, which is challenging in FNAB specimens [9]. 

Computed tomography (CT) is one of the most widely used imaging modalities to guide transabdominal biopsy. However, there has been considerable interest in the US in recent years because it offers high flexibility in the biopsy approach (transabdominal, transvaginal, and transrectal) and is a fast procedure [3]. Furthermore, the absence of radiation is an additional advantage over the use of the CT guide. 

While tru-cut in breast and prostate pathology is widely reported in the literature, much less is found in the gynecologic field. The works made by Fischerova [3] and Zikan et al. [10] are good examples of the latter. Both studies involved patients with an abdominopelvic mass with primarily inoperable pelvic tumors, poor performance status, and recurrent disease requiring histologic verification. The method was also applicable in gynecological patients for the technique’s safety, the preparations’ adequacy (93%), and the diagnostic accuracy (94.8–97.7%). Further publications confirmed even higher adequacy and accuracy of up to 100% [6,11]. Recently, Verschuere et al. [12] pointed out that diagnostic accuracy increased if at least two cylinders of tissue were available. 

This technique was not limited to oncological pathologies but was applied to diagnose benign pathologies, such as fibroids (24) and adenomyosis [13], playing a crucial role when dealing with patients suspected of oncology conditions. Moreover, the US-guided TCNB is a helpful tool in the diagnostic management of some particular cases of oncological patients [6], reducing diagnosis and treatment times but requiring US expertise and manual skills, and a deep knowledge of the pelvic and abdominal anatomy. 

One of our specific goals in this work is to provide additional clinical experience as evidence of the method’s feasibility and underline that, with an expert operator, the method is feasible in daily clinical practice, respecting the correct clinical indications. Nevertheless, the cases should be carefully selected to avoid the risk of spread and to provide the opportunity for cytoreduction in still eligible patients. 

The objective of this study is to prove the feasibility of adequacy, accuracy, and safety of this minimally invasive method of US-guided TCNB in the diagnostic management of the malignant or benign pelvic and abdominal masses in our center, to promote this technique as a routine procedure in a very selected population in the daily practice. 

## 2. Material and Methods

This retrospective study was conducted at the Gynaecological Department of the University of Trieste, Institute for Maternal and Child Health, IRCCS “Burlo Garofolo”. This procedure was introduced in our center in 2017 when the patient enrollments began. All patients who underwent a transvaginal or transabdominal TCNB between August 2017 and November 2021 were included. Patients or their legally authorized representatives were prior to inclusion in the retrospective study and gave their informed consent to the use of personal information. The Institutional Review Board of the Institute for Maternal and Child Health IRCCS Burlo Garofolo, Trieste, Italy (IRB-BURLO 02/2020 15 April 2020) approved this study, and 42 subjects were evaluated, and their clinical and US information were retrieved from the electronic medical records. We considered the following conditions for TCNB poor performance status unsuitable for surgery, suspected unresectable advanced tumors, inoperable advanced tumors (bulky malignant mass, diffuse visceral carcinomatosis, multiple, and unresectable metastases), suspicion of recurrence or metastases, and others (in particular, cervical tumors with unclear histology at vaginal biopsy). Patients scheduled for surgery were excluded from the study. All subjects underwent appropriate coagulation tests before TCNB and gave their written informed consent for the procedure. None of the enrolled patients presented contraindications to the procedure (thrombocytopenia, hemorrhagic diathesis, anticoagulant therapy if not suspended). Adequacy, accuracy, and safety were calculated based on the following:Adequacy was defined as the ability to obtain an amount of tissue able to determine the origin of the tumor and the ability to perform immunohistochemistry;Accuracy was classified as the concordance between the tru-cut needle ample and the postoperative histology;Safety was assessed based on complications rate.

### 2.1. US-Guided Tru-Cut Biopsy

All patients underwent both transvaginal and transabdominal US. A detailed report of the pelvis and abdomen by the US expert operator with more than 25 years of experience in the gynecological US was performed using a Voluson E6 (General Electric Healthcare GE, Zipf, Austria), with broadband from 5–9 MHz endocavitary transducer and 1–5 MHz transabdominal transducer. Power Doppler examinations were conducted at the pulse repetition frequency of 0.3 to 0.9 kHz, with the gain and pulse repetition frequency adjusted to define the vessels clearly. Lesions were described according to IOTA (International Ovarian Tumor Analysis) and MUSA (Morphological Uterus Sonographic Assessment) terms and definitions and categorized according to the “subjective evaluation” [14,15]. 

Solid, unilocular-solid or multilocular-solid lesions with sufficient solid components of at least 15 mm or more to allow tissue sampling were eligible for the procedure. 

The TCNB technique was employed using a fast-gun automatic biopsy system (Bard Magnum, Tempe, AZ, USA) for sampling with 16–18 Gauge tru-cut needles, G/16–30 cm core-cut biopsy needles. The biopsies were conducted with transvaginal or transabdominal US technique depending on the lesion site. Small lesions (less than 30 mm) adhering to large vessels, lesions accessible only transabdominal due to vaginal stenosis, outside the pelvis, or located significantly below the intestinal loops were not considered eligible for the procedure. The transrectal approach was possible but never performed. The cut penetration depth was controlled by setting the stopper at 15 or 22 mm depending on the lesion size, and adequate disinfection was executed with 10% povidone-iodine or chlorhexidine. Paracervical anesthesia was employed prior to transvaginal sampling when required by the patient, while local anesthesia with 10% mepivacaine was utilized in the abdominal wall in the transabdominal approach. The transvaginal procedure was performed in the lithotomy position, with a needle guide attached to the vaginal US probe, with an empty bladder after the vagina and anus disinfection. The transabdominal biopsy was conducted in the supine position, freehand under US needle tip guidance, paying close attention to the surrounding anatomical structures, avoiding necrotic tissue or the most vascularized parts. Figure 1 shows a typical example of this practice conducted in our institute, the US features of a cervical cancer relapse (left) and the US-guided TCNB of the same lesion (right). Samples were extracted from the potentially malignant mass, parietal carcinomatosis, metastatic lesions, or omental cake to obtain a cylindrical 15–22 long and 1–2 mm wide tissue for pathological examination. During the same procedure, we collected 2 or 3 samples from each target lesion, depending on the diameter of the lesion. A single sample was obtained in only one case because the lesion was too small to be determined with multiple sampling.

After the procedure, patients were checked for bleeding from the biopsy site, observed for a couple of hours in a dedicated area, and sent home if healthy.

In the case of bleeding from the sample location, we performed a tamponade lasting a few minutes until remission. Minor or major complications were reported if they occurred. 

### 2.2. Histopathological Diagnosis

The samples were fixed in formalin, embedded in paraffin wax, stained with hematoxylin-eosin, and then examined by a pathologist with more than 20 years of experience in gynecological oncology. The pathological results were available within 4–5 days to manage as soon as possible the subsequent treatments (chemotherapy, radiotherapy, palliative therapy, or surgery), decided with a multidisciplinary team approach, considering the overall clinical picture of the patient (i.e., age and comorbidities).

## 3. Statistical Analysis

Descriptive statistics were reported as median (range: min–max) for continuous variables or frequencies (percentages) for categorical variables. The TCNB accuracy in characterizing pelvic masses was assessed as the agreement between tru-cut biopsy histology and final operational histology in surgery patients.

## 4. Results

Over four years, 43 US-guided TCNB were performed on 42 patients. Table 1 summarizes the characteristics of the study population. Fifteen women (35.7%) had a history of gynecological cancer, while one patient had a previous colorectal cancer. Indications for the procedure were: inoperable advanced tumors (23 cases, 54.8%), poor health status (16 patients, 38%), suspicion of recurrence (11 patients, 26.1%) or metastases (3 patients, 7.1%), and previously inconclusive pathological exams obtained in one case (2.3%) by vaginal non-US-guided biopsy in suspected cervical cancer. Some patients had more than one indication for the procedure. 

### 4.1. Ultrasound Examination and Ultrasound-Guided Tru-Cut Biopsies

Table 2 reports the ultrasound characteristics of the lesions, the access, and the biopsy site. The suspicion of malignancy was present in all US-examined cases except for two, in which we determined their presumed benign nature (subsequently confirmed by tru-cut histology and long-term follow-up, which revealed the immutability of the lesion in time). In the first benign case, a young patient’s personal history of cervical cancer revealed benignity in the US-detected intracervical stromal lesion, which resulted later negative for neoplasm. In the second case, an obese patient with ascites and ovarian neoplasm (suspected of Meig’s syndrome) refused surgery for high intraoperative risk but accepted the outpatient TCNB.

Only one mild complication related to the procedure was reported. It involved a subject with severe hypertension who developed a nearly 8 cm endopelvic clot after the procedure, which resolved spontaneously. In the case of a massive hemorrhage, reported rarely and mainly in patients with hemorrhagic diathesis [10] (Zikan 2010), the major surgical approach is indicated.

### 4.2. Pathological Study

All specimens were classified as adequate for the histopathological study by the pathologist. In particular, one sample was repeated during a succeeding procedure in the suspect of recurrence of ovarian cancer and considered adequate. At the same time, another one underwent the US for massive bleeding in the strong suspicion of nearly 49 mm cervical stromal cancer. The patient repeated the procedure because, although three tissue samples were extracted from the lesion core, the pathological exam revealed a CIN3 in all three specimens. Considering the strong US suspicion pointing to cervical cancer, despite the negativity of the histological exam, a repeated TCNB was performed, revealing a focal keratinizing G3 solid infiltrating squamous carcinoma of the cervix. 

Table 2 summarizes the pathological exams of malign masses. In the case of advanced cervical tumors, the tru-cut was performed because no visible lesion at the speculum was present or a previous negative biopsy of the lesion. 

Sixteen patients underwent surgery (interval debulking surgery, recurrent disease, metastases) as foreseen by their therapeutic management. One case already described was followed for 3 years, and the lesion did not change for morphology or dimensions, which confirmed the benign nature of the lesion and therefore was considered benign in all respects. 

The final histology agreed with tru-cut biopsy diagnosis in all but two cases (diagnostic accuracy 88.2%), or if we consider only cases with at least two TCNB samples, diagnostic accuracy rose to 94.1%.

In one patient who underwent debulking surgery for a poorly differentiated endometrioid ovarian carcinoma, the TCBN histology revealed a peritoneal recurrence of high-grade serous ovarian carcinoma. Subsequent surgery proved a recurrence of endometrioid ovarian carcinoma. 

The leading cause of the difference between the two histological diagnoses can sometimes be the tumor’s heterogenicity. As happened in our case of CIN3, the first TCNB-based histology was performed on a lesion 49 millimeters in diameter, in which a US scan indicated its invasive nature. A second TCNB was conducted with a different yield, which demonstrated the challenge in determining the malignant histotype based on a small sample, as reported elsewhere for particular neoplasms [16]. 

## 5. Cases of Primary Interest and Discussion

A 52-year-old obese patient (BMI 35) underwent hysterectomy and bilateral salpingo-oophorectomy due to stage IV endometriosis 16 years earlier. She presented with pelvic pain, exertional dyspnea, weight loss, constipation, and a severe picture of thrombophlebitis of the right lower limb. The pelvic US reported an 18 cm multilocular-solid pelvic lesion with a color score of 3. Papillary vascularized projections were present within the mass, and multiple nodules involved the peritoneum (Figure 2). No ascites were present, but the omental cake was distinctly identifiable. Bilateral grade II hydroureteronephrosis was also reported. US scan by an expert operator helped to recognize that the lesion appeared to have similarities with the clear cell carcinoma of the ovary as a possible expression of malignant degeneration of an endometriotic remnant, although the adnexa was missing, as recently described elsewhere [17,18,19]. A US-guided transvaginal TCNB was performed due to the poor clinical condition of this woman. The pathological exam diagnosed a high-grade clear cell adenocarcinoma of the ovary (immunohistochemistry: CK7+, CK20−, Napsin+, PAX-8+, Vimentin+), with the ascertained endometriotic origin of the lesion. Therefore, it was decided to continue with chemotherapy with paclitaxel and carboplatin. This case demonstrated that pelvic or abdominal endometriotic lesions might undergo malignant degeneration. Recently, the literature highlighted the possible postmenopausal malignant degeneration of endometriotic lesions already present [18]. Nevertheless, it is essential to highlight that the lesions may arise as an expression of malignant degeneration even in patients who have undergone a complete hystero-adnexectomy at surgery, as observed in our case.

### 5.1. Non-Gynaecological Masses

We draw attention to the 3 non-gynecological cancers found during our study: 2 patients with ovarian and colorectal cancer metastasis and 1 with pelvic site type B lymphoma metastasis. The first two cases had typical ultrasound characteristics of the underlying pathology [20,21,22]. The following TCNB confirmation allowed planning a tailored therapy for the patients in a short time. 

Concerning the third case (the type B lymphoma), the subject had already undergone multiple diagnostic investigations looking for the origin of the tumor. In particular, she was referred for the suspicion of cervical cancer due to a positron emission tomography/CT image suggestive of a cervical cancer lesion. The uterus showed no anomalies in clinical and US exams, while a highly vascularized retropubic lesion was observed during the US scan (Figure 3). TCNB revealed the presence of a B lymphoma pelvic metastasis after the histology, and the patient started chemo-immunological therapy at once. This case demonstrated how TCNB, if performed by an expert operator, is crucial for a rapid and correct diagnosis with minimal invasiveness, which corroborates with previous reports on the validity of the method and role of US-guided TCNB in detecting neoplastic relapses [3,6,10,23].

### 5.2. TCBN in Recurrent Cancer

The advantage of using the US in the follow-up of gynecological cancers has been demonstrated since 1987 [24,25,26,27]. Testa et al. [27] stated that the positive predictive value of the US examination in detecting the recurrence of gynecological malignancies was 100%, while the negative one was 92.7%. Its role is crucial in recognizing recurrences and their biopsies as a guide to obtaining a histological diagnosis. The diagnosis of recurrent ovarian cancer after debulking surgery through US-guided TCNB was first described by Fischerova et al. [3]. They reported a very low complication rate (<1%), an accuracy of 98%, and adequacy of 95%. 

The US-guided TCNB is a valuable method for those patients who underwent ultrasound follow-up positive for recurrence after debulking surgery. The technique is more effective if the lesion involves the pelvic region, and its accuracy is so high that it provides helpful information on the therapeutic route, avoiding unnecessary surgeries [6,28]. The US guidance leads the probe to be positioned with the correct angle and pressure, moving the bowel loops, avoiding excessive bleeding, and allowing sampling at the most appropriate site of the lesion in an outpatient setting. 

### 5.3. TBCN: Our False-Negative Results

In this picture, the TCBN was not useful for diagnosis on two occasions. We dealt with a small-sized lesion from a poor health and cooperating woman in one case. The decision to carry out the procedure was shared with the awareness that results would not change the poor prognosis. Only one specimen was obtained from the parietal carcinomatosis in the Douglas pouch during the vaginal procedure of the target disease. In a presumed clinical picture of inoperability, the principal US imaging (Figure 4a–d) highlighted the presence of ascites, diffuse pelvic, abdominal parietal, visceral carcinomatosis (a), omental cake (b), mesenteric radix (c), and a nodule of carcinomatosis on the descending colon (d). In retrospect, perhaps, the choice of the transabdominal access at the omentum level would have provided better results. Accordingly, the transvaginal route was usually preferred for the best resolution, the absence of overlapping bowel loops, and the possibility of reducing any bleeding with pressure [12]. In addition, in previous work [10], this approach produced better results in terms of adequacy, but, as shown by our images below (Figure 4a–d), the scarceness of the tissue seen in the US (Figure 4a) led us to choose another site easily accessible as omentum.

The abdominal CT scan confirmed the same US report, while the thoracic CT scan revealed pleural effusion and multiple thoracic metastatic lesions associated with hilar lymphadenopathy. Before the TCNB procedure, the patient underwent evacuative paracentesis to alleviate respiratory symptoms. Cytological examination proved the presence of abundant malignant neoplastic cells of uncertain origin. The decision to perform the method or not was established with the patient explaining that the pathology was pervasive, and that the prognosis was quite unstable. The patient agreed to perform the method in full consciousness. It is not uncommon to face a quite advanced clinical oncological setting for which the prognosis remains rather challenging, although it is possible to understand quickly the nature of the lesion using TCNB. This procedure was proposed to the patient and her family to unveil the disease’s nature as non-invasive and quick. Although they were told that the prognosis did not change, the knowledge of the cause could reassure them, and in full consciousness, the patient chose to investigate.

In the second case, the sample was obtained from a very small neoformation of the maximum longitudinal diameter of 18 mm (volume of 2.5 cc) near the vaginal dome. The tissue on the vaginal dome had the macroscopic appearance of being necrotic, so a TCNB was preferred on the assumption that this technique could sample more in-depth into the lesion. However, as shown in Figure 5, this small lesion developed mainly towards the vagina. Due to a negative TCNB result, a subsequent vaginal biopsy with forceps was performed, resulting in an endometrial endometrioid adenocarcinoma recurrence. 

As already known from the literature, there is a high likelihood of success of the TCNB when target lesions have more than 2 cm in diameter [23]. Some other TCNB favorable predictors should be further discussed because there are several pieces of evidence. Zikan et al. [10] analyzed 195 women undergoing US-guided TCNB: the presence of ascites, elevated CA-125, suboptimal operable primary tumor, serum ovarian carcinoma, carcinomatosis, and vaginal approach were more likely associated with a successful procedure. Others [23] reported that the final result was not influenced by the histological type of the tumor, its vascularity, and the presence of ascites. On the other hand, possible failure predictors are low dimensions of the target lesion, inadequate site of the TCNB, frozen section procedure, and interpretation errors [29]. 

According to others [6], we should be focused on the biopsy site to prevent inadequate preparation; cystic or necrotic portions should be avoided. The superiority of the technique over blind biopsies is usually evident for recurrences of endometrial cancer over the vaginal dome or cervical tumors located in the upper part of the cervix. The US-guided TCNB identifies the most suitable area for biopsy, thus improving the histological results’ adequacy and accuracy.

It has become a consensus that the expert US obstetrics and gynecology specialist must be able to identify the most suitable portion of tissue for biopsy to obtain the appropriate result. Sampling errors can be easily avoided by being careful to obtain enough material [3] and bypassing sampling in the cystic or necrotic site of the mass, as cellular presence is insufficient for diagnosis. A dedicated expert sonographer is also essential. Subjective US assessment and diagnostic algorithms (ADNEX model) [11,30] can help establish the origin and the nature of the lesion before TCNB.

### 5.4. TBCN in Advanced Ovarian Cancer

Ten patients in an advanced stage of pathology with TBCN positive for ovarian cancer underwent appropriate neoadjuvant chemotherapy and subsequent cytoreductive surgery, saving time for the beginning of treatment and avoiding surgery which, in addition to the need for general anesthesia and increased costs, can potentially cause intraoperative morbidity and the onset of port-site metastasis in up to 17–49% [12,31,32,33].

Only one patient had urgent surgery because of a bowel occlusion that required an ileostomy and diffuse lysis of adhesions. Part of the omentum involved in the disease was sent for a pathological exam. The final pathological exam (of target lesions and involved omentum) agreed with the TCNB results in all these cases. 

### 5.5. TCBN in Benign Masses

In this study, only two cases were representative of benign tumors. In the first case, the procedure involved a 39-year-old woman who had previously undergone conization and systematic pelvic lymphadenectomy due to a G2 invasive squamous carcinoma of the cervix three years earlier. The US scan detected a round, solid, myometrial lesion with regular margins and a color score of 2 of about 15 mm. In this case, the TCNB was necessary to confirm that the lesion was not a recurrence of the disease. The final examination confirmed the presence of a benign smooth muscle cell lesion. 

The second case involved a 51-year-old woman, highly obese (BMI 42.5) with multiple comorbidities, complaining about a sudden onset of pelvic pain. The abdominopelvic US revealed the presence of ascites associated with an ovarian solid rounded, shadowing mass with a color score of 2, described according to IOTA (International Ovarian Tumor Analysis) terms and definitions [14], suggestive of a type of ovarian fibroid of 12 cm in size (Meig’s syndrome). The surgical risk was high, and for that reason, the patient refused it. Despite knowing diagnostic limitations related to the size of the mass, it was decided, in agreement with the patient, informed of the possible limits, to perform a TCNB on the lesion, which confirmed the benign diagnosis. After a few weeks, the patient underwent major surgery for worsened respiratory difficulties due to ascites, and the diagnosis of Meig’s syndrome was confirmed. 

Considering that TCNB is an efficient diagnostic technique, it can also be helpful in suspected benign diseases, according to some authors [34]. El Hachem et al. [34] reported the potential TCNB role in the intraoperative assessment of presumed fibroids before power morcellation, and Kawamura et al. [35] observed them as a preoperative tool in the differential diagnosis of uterine myoma-like tumors, given the low risk of dissemination of potentially malignant cells [23]. However, Zikan, in 2010, described the only two cases in which the final histology was not consistent with the final histological TBCN results. One of these referred to a diagnosis of benignity in a lesion demonstrated to be definitively a low-grade leiomyosarcoma. The author pointed out that in this type of tumor with lower proliferation activity, the TCNB sample, especially if done in the periphery of the lesion, might, in many cases not be diagnostic [10].

In our 4 years of experience, we retrospectively analyzed the reliability in terms of adequacy, accuracy, and safety of US-guided TCNB in the presence of pelvic and extra pelvic primary or metastatic lesions. The TCNB provided adequate samples for histological analysis in all cases (100%) with high diagnostic accuracy (94.1%) if at least two samples were collected. Despite the low sample number and the study’s retrospective nature, which undoubtedly limited its results, our findings complied with the literature and confirmed the methodology. The adequacy of the technique in the literature was reported between 85 and 100%, while diagnostic accuracy was between 95–100% depending on the site and origin of the lesion [1,3,6,8,10,11,12,23,29,36,37]. From the results calculated on 94 cases by Lengyel et al., sensitivity was 94.8%, specificity 94.1%, positive predictive value 98.6%, and negative predictive value 80.1% [37]. The diagnostic inaccuracy between the histological outcome of TCNB and the final histological examination seemed to reach 12.8% in the literature and was attributable to differential diagnostic difficulties, primarily due to tumor heterogeneity and sampling errors [37]. Nevertheless, according to the authors, the diagnostic inaccuracies found did not adversely affect the management and treatment of these patients, as the TCNB had correctly identified the dignity of these lesions [37].

Due to the high performance of the procedure, the TCNB technique has been recently introduced in the primary workup of cervical cancer to reduce false-positive findings by imaging methods and thus avoid inappropriate treatment [38] and false-negative results. As demonstrated by Mascilini et al., some cervical tumors were not identified with colposcopy because the lesion did not involve the portio [6]. Indeed, the transvaginal approach allowed extreme precision, even in samples of deep lesions, not always easily accessible with other imaging methods. Rarely the multifocality of the lesion could require the execution of an additional procedure as described in our series. Nevertheless, the use of TCNB was not limited to cervical cancers: a selected and well-defined group of patients who were not suitable for surgery with an ovarian disease or advanced tumor disease benefited from this technique [12,23]. Approximately 30% of patients with ovarian cancer have a sign of inoperability at the time of diagnosis, often occurring in the advanced stage of the disease when a prompt histological diagnosis is required. However, the tru-cut diagnostic path in ovarian cancer patients is dedicated exclusively to those who result unsuitable for surgery due to advanced inoperable disease, poor health conditions, to whom a surgical procedure may be at too high risk, or to those who have already been treated for cancer and clinical or instrumental reasons appear to have a recurrence. It was recently reported in an ESGO/ISUOG/IOTA/ESGE consensus statement that preoperative imaging should not influence the choice of treatment of the patient with ovarian cancer concerning predicting peritoneal tumor resectability [39] and care must be taken not to perform TCNB in patients who could benefit from cytoreductive surgery or in those in whom there is a risk of disseminating the neoplastic pathology.

The US-guided TCNB can be considered an accurate diagnostic method suggested in selected cases. It is an inexpensive and feasible technique. However, its use in gynecology was not spread as in other anatomical sectors: it has a broader use in the breast field, probably because the target organ is easy to address. It could also be employed in other contexts, in which the use of this safe and straightforward technique allows obtaining prompt diagnosis directing patients to the best therapeutic path.

Some authors reported the investigation of pelvic or abdominal non-gynecological tumors [36], suggesting a role of the TCNB in other contexts (urological, hematological, or surgical) through other sites of access. For example, a transrectal approach could have enormous potential in the urological or surgical context. 

From our experience, in all cases, the sample obtained through the TCNB represents the disease and is suitable for histological and immunohistochemical analysis. The diagnostic accuracy of the US-guided TCNB is high due to the relative feasibility of the method performed by a dedicated US expert specialist. Nevertheless, it requires US expertise, manual skills, and perfect pelvic and abdominal anatomy knowledge.

## Figures and Tables

**Figure 1 jcm-11-02534-f001:**
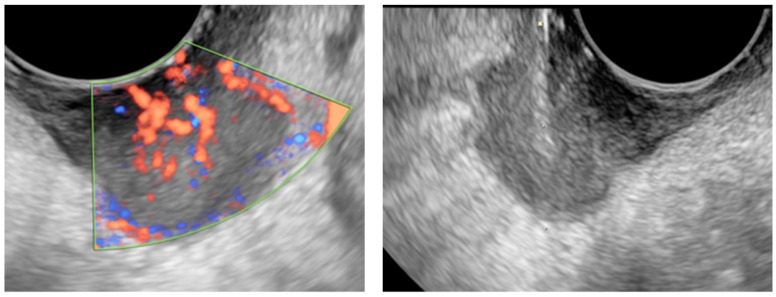
The US image of a highly vascularized cervical squamous cancer relapse (**left**) and the clear image of the tru-cut needle inside the lesion during the procedure (**right**).

**Figure 2 jcm-11-02534-f002:**
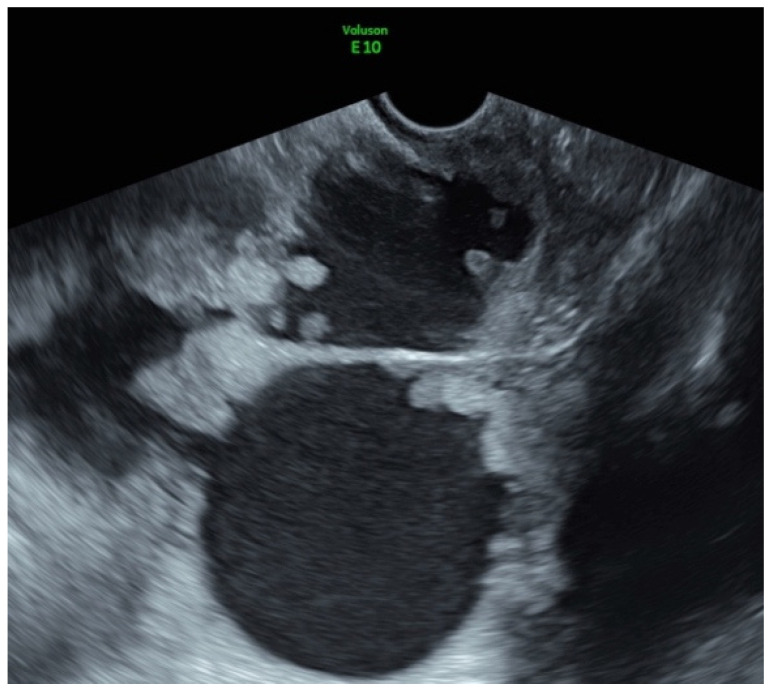
Transvaginal US picture of a multilocular-solid lesion with papillary projections (high-grade clear cell adenocarcinoma of the ovary).

**Figure 3 jcm-11-02534-f003:**
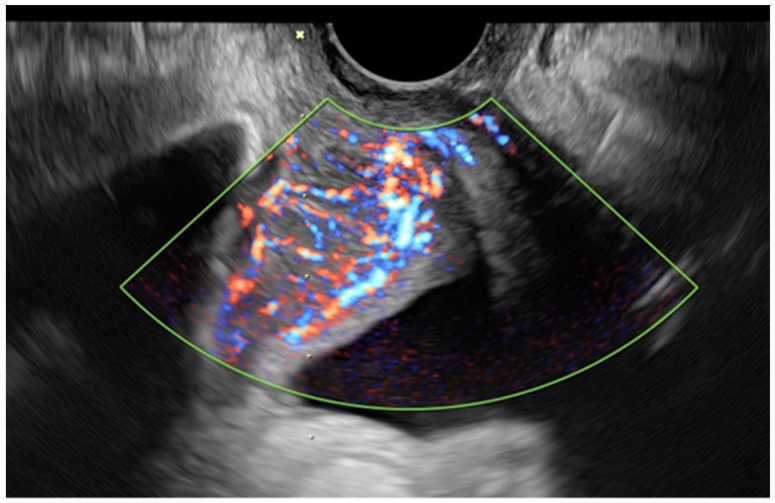
The US image of the pelvic B lymphoma shows a solid lesion with irregular and shaded margins and a color score of 4 between the urethra and the pubic bone.

**Figure 4 jcm-11-02534-f004:**
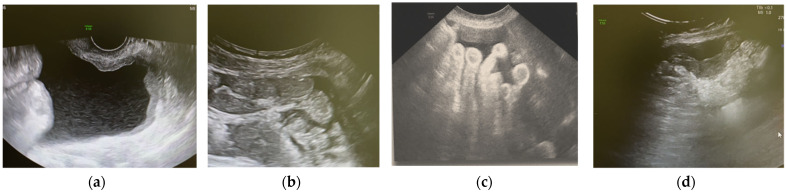
The US images of ascites, diffuse pelvic and abdominal parietal, and visceral carcinomatosis (**a**), omental cake (**b**), the presumed involvement of the mesenteric radix (**c**), a nodule of carcinomatosis on the descending colon (**d**) in a plausible clinical picture of inoperability.

**Figure 5 jcm-11-02534-f005:**
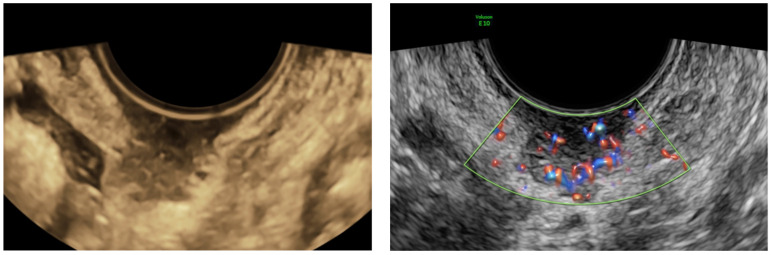
On the (**left**): 2D US images with VCI of a solid neoformation from the vaginal dome in a patient with a previous diagnosis of endometrial cancer. On the (**right**): Doppler US image of the same neoformation highlighting the vascularization in the lesion.

**Table 1 jcm-11-02534-t001:** Characteristics of the patients.

Variables	N = 42 Patients
Age (years): Median (min-max)	72 (39–93)
Body Mass Index (Kg/m^2^): Median (min-max)	24.9 (18.6–42.5)
CA-125 UI/L: Median (min-max)	129 (3–2358.3)
Personal history of cancer (N, %)	15 (35.7%)
*Indications to the TCNB*	
Inoperable advanced tumor (N, %)	23 (54.8%)
Poor performance status (N, %)	16 (38%)
Suspicion of recurrence (N, %)	11 (26.1%)
Suspicion of metastases	3 (7.1%)
Previously undefined malignancies (N, %)	1 (2.3%)

**Table 2 jcm-11-02534-t002:** US features and final pathology.

Variables	N = 42 Patients
Largest diameter of the lesion (mm): median (min-max)	51 (8–280)
*Type of the tumor (N, %)*	
Solid	34 (81%)
Multilocular-solid	8 (19 %)
*Tumor margins (N, %)*	
Irregular	36 (85.7%)
Regular	6 (14.3%)
*Color score (N, %)*	
2	10 (23.8%)
3	29 (69%)
4	3 (7.1%)
Ascites (N, %)	12 (28.6%)
*Site of access (N, %)*	
Transvaginal	31 (73.8%)
Transabdominal	11 (26.2%)
*Site of biopsy (N, %)*	
Lesion	34 (81%)
Omental cake	4 (10%)
Carcinosis	4 (10%)
*Histology (N, %)*	
Benign	2 (4.8%)
Malign	40 (95.2%)
Primary advanced tumors	27 (67.5%)
Advanced ovarian cancer	19 (47.5%)
Advanced cervical cancer	7 (17.5%)
Recurrent genital tumors	10 (25%)
Recurrence of endometrial cancer	4 (10%)
Recurrence of cervical cancer	3 (7.5%)
Recurrence of ovarian cancer	3 (7.5%)
Primary peritoneal cancer in an oncological patient (leiomyosarcoma)	1 (2.5%)
Metastases or non-genital malignancy	3 (7.5%)
Complications (mild) (N, %)	1 (2.4%)

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
