# Peer review of "Ultrasound-Guided Tru-Cut Biopsy in Gynecological and Non-Gynecological Pelvic Masses: A Single-Center Experience"

_jcm, 2022, doi:10.3390/jcm11092534_

Round 1

Reviewer 1 Report

In the present manuscript, ”Ultrasound-guided tru-cut biopsy in gynecological and non-gynecological pelvic masses: a single-center experience,” the authors investigated the use of ultrasound-guided biopsy. This study is well-written overall and clinically interesting. However, there were several minor concerns, and the detailed comments are described below.

  1. Ultrasound-guided biopsy is considered to be highly safe. However, in case of massive bleeding, how to manage it?
  2. Please clarify the characteristics of eligible tumors (solid or cystic, size, location, etc.).

Author Response

Dear,

We thank the reviewer for their constructive criticism, and time spent to analyze this manuscript. The responses, and explanations related to their comments are listed in the following:

  1. English can be improved

The text has been re-assessed and improved in grammar.

  1. Methods can be improved

To improve this paragraph of the text, we defined which tumours were considered eligible for the procedure (see text and monitoring question).

  1. Ultrasound-guided biopsy is considered to be highly safe. However, in case of massive bleeding, how to manage it

“Massive bleeding is a very rare complication of tru-cut biopsy that in our case series, although contained, has never occurred. Only one minor complication has occurred, which is the development of a small clot after the procedure. Since tumors are generally highly vascularized, this possibility needs to be considered. Other risk factors for bleeding that can make the procedure more or less safe need to be assessed first. For all patients, as already reported in the text, an observation of a few hours is expected to identify any complications and possible development of external bleeding or hematoma. Massive bleeding can be acute (if it develops a few hours after the procedure), resulting in the development of anemia, acute abdomen, and haemorrhagic shock, or chronic (even after weeks) due to continued blood spillage from a small vessel, subsequent progressive anemia, and development of a generally organized clot.

The treatment is therefore decidedly different: in acute cases, a laparoscopic or laparotomic surgical approach may be required to stop the bleeding and a massive transfusion, as well as cardiovascular support in cases of haemorrhagic shock. In the chronic stabilized forms, instead, strict monitoring and antibiotic prophylaxis may be preferred, resorting to surgery only if necessary.”

The text provides a summary of the management of the problem. Please let us know if the summary reported is adequate

  1. Please clarify the characteristics of eligible tumors (solid or cystic, size, location, etc.).

Solid or multilocular-solid lesions with sufficient solid component of at least mm15 or more to allow tissue sampling were considered eligible for the procedure.

The TCNB technique was performed using a fast‐gun automatic biopsy system (Bard Magnum, USA) for sampling with 16-18 Gauge tru-cut needles, G/16-30 cm core‐cut biopsy needle. The biopsies were conducted by transvaginal or transabdominal US technique depending on the lesion site. Small lesions (less than 30 mm) adhering to large vessels or lesions accessible only transabdominally because of vaginal stenosis or because outside the pelvis, and/or very deeply located below the intestinal loops, were not considered eligible for the procedure. The transrectal approach was possible but never performed.

Reviewer 2 Report

Please correct some corrections errors: vs should be written in italics, 0.9kHz should be 0.9 kHz, before the figure and table numbers there should be no dash.

The information in the text should not duplicate the information in the tables.

Merge the second table with the third.

The conclusion that "safety is high if the technique is performed by an experienced operator" should be corrected. It is not defined who an experienced operator is.

An explanation of what are minimally invasive biopsy methods would be needed.

Author Response

Dear,

We thank the reviewer for their constructive criticism, and time spent to analyze this manuscript. The responses, and explanations related to the comments are listed in the following:

1- Introduction can be improved

We reorganized the description of mini-invasive methods, which made the text clearer and fluent

2 – Results can be improved in clarity

The results have been made smoother by avoiding repetitions between text and tables as suggested. Furthermore, Tables 2 and 3 have been merged together.

3 - Please correct some corrections errors: vs should be written in italics, 0.9kHz should be 0.9 kHz, before the figure and table numbers there should be no dash.

We have corrected according to the suggestion of the reviewr

4 - The information in the text should not duplicate the information in the tables.

The data entered in the text and tables, if duplicated, have been removed, as suggested.

5- Merge the second table with the third.

The second and the third tables have been merged as requested.

6- "safety is high if the technique is performed by an experienced operator" should be corrected. It is not defined who an experienced operator is.

It was deleted in the section “conclusion” of the abstract

7 - An explanation of what are minimally invasive biopsy methods would be needed.

We reorganized the description of mini-invasive methods, which made the text clearer and fluent.